# Exploring the spatial variation and associated factors of childhood febrile illness among under-five children in Ethiopia: Geographically weighted regression analysis

**Negalgn Byadgie Gelaw**[1]*, **Getayeneh Antehunegn Tessema**[2], **Kassahun Alemu Gelaye**[3], **Zemenu Tadesse Tessema**[4], **Tigist Andargie Ferede**[5], **Abebe W/Selassie Tewelde**[6]

1 Department of Public Health, Mizan-Aman College of Health Sciences, Mizan-Aman, Ethiopia,
2 Department of Epidemiology and Biostatistics, College of Medicine and Health Sciences, University of Gondar, Gondar, Ethiopia, 3 Institute of Public Health, College of Medicine and Health Sciences, University of Gondar, Gondar, Ethiopia, 4 Department of Epidemiology and Biostatistics, College of Medicine and Health Sciences, University of Gondar, Gondar, Ethiopia, 5 Maternal Mortality Reduction and Child Health, Gondar, Amhara Regional State Ethiopia, 6 Department of Nursing, College of Medicine and Health Sciences, University of Gondar, Gondar, Ethiopia

* negalgnbyadgie21@gmail.com

## Abstract

### Background

The global burden of febrile illness and the contribution of many fever inducing pathogens have been difficult to quantify and characterize. However, in sub-Saharan Africa it is clear that febrile illness is a common cause of hospital admission, illness and death including in Ethiopia. Therefore the major aim of this study is to explore the spatial variation and associated factors of childhood febrile illness among under-five children in Ethiopia.

### Methods

This study were based on the 2016 Ethiopian Demographic health survey data. A total weighted sample of 10,127 under- five children was included. Data management was done using Stata version-14, Arc-GIS version—10.8 and SatsScan version- 9.6 statistical software. Multi-level log binomial model was fitted to identify factors associated with childhood febrile illness. Variables with a p-value < 0.2 in the bi-variable analysis were considered for the multivariable analysis. In the multivariable multilevel log binomial regression analysis p-value< 0.05, the APR with the 95% CI was reported. Global spatial autocorrelation was done to assess the spatial pattern of childhood febrile illness. Spatial regression was done to identify factors associated with the spatial variations of childhood febrile illness and model comparison was based on adjusted R2 and AICc.

### Result

The prevalence of febrile illness among under-five children was 13.6% (95% CI: 12.6%, 14 .7%) with significant spatial variation across regions of Ethiopia with Moran's I value of

**Data Availability Statement:** All relevant data are within the paper and its Supporting Information files.

**Funding:** The author(s) received no specific funding for this work.

**Competing interests:** The authors have declared that no competing interests exist.

**Abbreviations:** AICc, Corrected Akaki Information Criteria; Df, Degree of Freedom; EA, Enumeration Areas; GWR, Geographically Weighted Regression; LL, Log Likelihood; NGO, Non-Governmental Organization; OLS, Ordinary Least Square; PNC, Postnatal Care; UOG, University Of Gondar; VIF, Variance Inflation Factor; ANC, Antenatal Care; APR, Adjusted Prevalence Ratio; ARI, Acute Respiratory Infection; BIC, Bayesian Information Criteria; CV, Community Variance; MGWR, Multi-scale Geographically weighted regression.

0.148. The significant hotspot areas of childhood febrile illness were identified in the Tigray, Southeast of Amhara, and North SNPPR. In the GWR analysis, the proportion of PNC, children who had diarrhea, ARI, being 1st birth order, were significant explanatory variables. In the multilevel log binomial regression age of children 7–24 months(APR = 1.33, 95% CI: (1.03, 1.72)), maternal age 30–39 years (APR = 1.36 95% CI: 1.02, 1.80)), number of children (APR = 1.78, 95% CI: 0.96, 3.3), diarrhea(APR = 5.3% 95% CI: (4.09, 6.06)), ARI (APR = 11.5, 95% CI: (9.2, 14.2)) and stunting(APR = 1.21; 95% CI: (0.98, 1.49) were significantly associated with childhood febrile illness.

## Conclusion

Childhood febrile illness remains public health problem in Ethiopia. On spatial regression analysis proportion of women who had PNC, proportion of children who had diarrhea, proportion of children who had ARI, and proportion of children who had being 1st birth order were associated factors. The detailed map of childhood febrile illness and its predictors could assist health program planners and policy makers to design targeted public health interventions for febrile illness.

## Background

Childhood febrile illness is a common cause of hospital admission, and its associated infectious causes contribute to substantial morbidity and death among children worldwide, especially in low- and middle-income countries including Ethiopia [1].

Although considerable progress has been made in reducing mortality from 93 to 39 deaths per 1000 live births between 1990 and 2018, this remains a major public health issue [2].Out of all causes, infectious diseases have been responsible for the greatest global burden of death and disability among children under 5 years of age.

Many of the challenges for estimating the burden of febrile illnesses could be addressed by reorganizing diseases presenting predominantly with fever into a syndrome "envelope" of febrile illness [3]. Undifferentiated fever is the main clinical feature of many diseases of global importance, including malaria, invasive bacterial diseases [4]. The causes of severe acute febrile illness are numerous, and account for many preventable deaths in low income countries, particularly in children and is most frequently due to infection [5]. The global burden of febrile illness and the contribution of many fever-inducing pathogens have been difficult to quantify and characterize. However, in sub-Saharan Africa it is common [4], and febrile illness is a common cause of death [6]. In Africa, 35.7% of all self-reported fevers were accompanied by a malaria infection in 2014, but that only 28.0% of those were causally attributable to malaria [7]. There is difficulty in identifying African children in need of antibiotics among the majority who do not [8]. Differences in the patients' demographic and clinical characteristics could affect the intervention programs for treating febrile illness [9]. As a result, the effectiveness of diagnosis and treatment was questionable.

Although massive investment in diseases responsible for fever among children less than five years in Ethiopia, the burden of the febrile illness remains high and disproportionately distributed across regions [10]. In Ethiopia, studies assessing the spatial variation and associated factors of childhood febrile illness, especially at a local scale had limited evidence. Even though all population affected by febrile illness, children are amongst the most vulnerable to febrile illness that to deaths in the country [11].

The associated factors for childhood febrile illness previously documented research in Bangladesh, 31% of children aged less than five years die from febrile illness [12], age of children, number of children at home and wealth index [13], child advances in age, educational status [14], area of residence, wealth index for febrile illness [15], having a cough in the last 2 weeks, region [16], diarrhea, and acute respiratory infections [17]. Un-improved WASH conditions risk factor febrile illness among Nigerian children [18]. Malnutrition like underweight, wasting and stunting [19], Latrine type [20]. In Ethiopia first birth order, non-exposure to mass media, maternal age <30 years, absence of antenatal and postnatal care utilization, and being born from uneducated mothers were considered as significant determinant factor [13] sharing the house with livestock increased the risk [21], vaccination coverage [22] Children living in poor and unsanitary environments, younger children, and those sick, stunted, wasted, or consuming diets low in vitamin A [23], Marital status and giving birth at health facilities [24] were significant factors of child hood febrile illness.

The government of Ethiopia, was given more attention for maternal and child health care, the burden of the febrile illness in children remains high and disproportionately distributed across region. According to studies reported previously on spatial distribution and associated factors of febrile illness, the prevalence of child hood febrile illness has variation across regions in Ethiopia [25].

Spatial techniques also help to identify hotspots area of febrile illness and provide information that enables public health programmers and policymakers in strategic planning [26] Local estimates of the febrile illness burden can be used to prioritize febrile illness care and prevention interventions among marginalized populations living in remote or conflict areas of Ethiopia.

Therefore, this study aimed to explore the spatial distribution childhood febrile illness and its associated factors among under -five children in Ethiopia using GWR analysis. The findings of this study will help for policymakers, health professionals, and other health care programs like NGOs, in guiding health programs and prioritize prevention and intervention programs. In addition, identifying hotspot areas of childhood febrile illness by map will provide a deeper understanding of the impacts of current implemented interventions in each region of the country.

## Methods and materials

### Study area, data source, and study period

This study was based on the 2016 Ethiopian Demographic and Health Survey data. The 2016 EDHS was the fourth DHS in Ethiopia, which was conducted every 5 years. The EDHS is mainly aimed to generate updated health and health-related indicators such as maternal mortality, child mortality, family planning, vaccination, and maternal health care service utilization. Ethiopia is administratively divided into nine geographical regions (Tigray, Afar, Amhara, Oromia, Somalia, Benishangul-Gumuz, Southern Nation Nationality, and People's Region (SNNPR), Gambella, and Harari) and two self-administrative cities (Addis Ababa and Dire Dawa) [27]. Each region is subdivided into zones, each zone into whereas, and each Wereda divided into Kebeles. A multistage stratified sampling technique was applied to select the study designs. In the first Enumeration Areas was randomly selected and in the second on average 28 households per clusters/EAs.

### Population

All under-five children in Ethiopia within the five years preceding the survey were the source population and All under-five children in Ethiopia who were in the selected clusters were

considered as the study population, Children in Ethiopia who were in the selected clusters that complete data information were included. Children with incomplete data, and longitude and latitude zero degree for spatial data exploration was excluded from the analysis. The EDHS has several datasets such as men (MR file), women (IR file), children (KR file), birth (BR file), and household (HR file) datasets. For this study, we used the Kids Record dataset (KR file), and a total weighted sample of 10, 127 under five children was included.

## Study variables

The dependent variable was the febrile illness prior to 2 weeks duration in kid record status of children under five years, which was categorized into yes coded as "1" and no coded as "0". The independent variables considered in this study were age of children, sex of children, type of birth, number of children, Vitamin A in the last 6 months, ARI, Diarrhea, vaccination, birth order, region, residence, maternal age, maternal educational status, ANC, PNC household wealth status, media exposure, type of birth, water source, type of toilet facility, wasting status, underweight status and stunting status. Wasting; children with weight-for-height Z-score <-2standard deviation. Stunting children with height-for-age Z-score <-2standard deviation. Underweight; children with weight-for-age Z-score <-2standard deviation.

## Data management and analysis

Data extraction, coding, and analysis were done using Excel, SatScan, Stata version 14 and, Arc-GIS version-10.8 and SatScan version 9.6 statistical software. The weighted data were used for analysis to restore the representativeness of the data. Since the EDHS data has a hierarchical nature, the Intra-class Correlation Coefficient was estimated to assess the clustering effect. The ICC indicated that there was a clustering effect intra class correlation coefficient >10%. Multi-level log binomial model was fitted to identify factors associated with childhood febrile illness. Variables with a p-value < 0.2 in the bi-variable analysis were considered for the multivariable analysis. In the multivariable multilevel log binomial regression analysis, the APR with the 95% CI was reported. Global spatial autocorrelation was done to assess the spatial pattern of childhood febrile illness. Spatial regression was done to identify factors associated with the spatial variations of childhood febrile illness and model comparison was based on adjusted R2 and AICc.

## Spatial analysis

The global spatial autocorrelation was done to assess whether the spatial distribution of childhood febrile illness among under five children in Ethiopia was dispersed, clustered, or randomly distributed [28]. Global Moran's I is a spatial statistic used to measure spatial autocorrelation by taking the entire data set and produce a single output value that ranges from -1 to +1. Moran's, I value close to −1 indicates that febrile illness among under-five children is dispersed, whereas Moran's I close to +1 indicates febrile illness among under-five children is clustered and if Moran's I close to 0 revealed that febrile illness among under-five children is randomly distributed. A statistically significant Moran's I when p-value < 0.05 value showed that febrile illness among under-five children is non-random. The hotspot analysis was done using the Getis-OrdGi* statistics to explore how spatial autocorrelation varies over the study location by calculating GI* statistic for each area. Z-score is computed to determine the statistical significance of clustering, and the p-value is computed for the significance. Statistical output with high GI* indicates "hotspot" whereas low GI* means a "cold spot" [29].

## Spatial regression analysis

The Ordinary Least square regression and Geographic Weighted Regression statistical analysis were employed for exploring the spatial relationship between febrile illness among under-five children and the explanatory variables, which affect the dependent variable. The outcome variable for spatial regression analysis was the percentage of febrile illness among under-five children at the EA level. A neighborhood or bandwidth is the distance band or the number of neighbors used for each regression equation; it is the most important parameter for spatial regression as it controls the degree of smoothening in the model. The complexity of spatial regression model depends not only by the number of variables in the model but also the bandwidth. There are three choices of band width methods such as AICc, CV and bandwidth parameter. For this study we have used adaptive kernel whose bandwidth was found by minimizing the AICc value.

**Ordinary least squares regression.**   The spatial regression modeling was performed to identify predictors of the spatial heterogeneity of febrile illness among under-five children. OLS is a global statistical model for testing and explaining the relationship between the dependent and independent variables [36]. It uses a single equation to estimate the relationship between the dependent and independent variables and assumes stationarity or consistent relationship across the study area. The OLS was used as a diagnostic tool and for selecting the appropriate predictors and uses for the Geographic weighted regression model [30].

The OLS can automatically check the multi-collinearity between independent variables. The multi-collinerity was assessed using the Variance Inflation Factor. If the VIF values are greater than 7.5 in the OLS model, it indicates the existence of multi-collinearity among the explanatory variables and should apply to leave one out an approach based on the VIF values. In addition, the autocorrelation statistic was applied to detect whether there is spatial autocorrelation or clustering of the residuals which violates the assumptions of OLS. The spatial independence of the residuals was assessed with the global spatial autocorrelation coefficient Moran's I value to asses GWR done or not.

## Geographically weighted regression

A local spatial statistical technique that assumes the non-stationary in relationships/ heterogeneity in the relationship between the dependent and explanatory variables across EAs The GWR analysis is considered when the Koenker statistics is significant when p-value<0.05, which means that the relationships between the dependent and the independent variable change from location to location. In the GWR analysis, the coefficients of the explanatory variables take different values across the study area. Mapping the GWR coefficients associated with the explanatory variables, which are produced using the GWR, provides insight for targeted interventions. In our study we use the corrected Akaike Information Criteria and adjusted R-squared for model comparison of OLS and GWR model. A model with the lowest AICc value and a higher adjusted R-squared value was considered as the best-fitted model for the data.

## Multi-scale geographically weighted regression

In multi-scale Geographical weighted regression assumes the non-stationary in relationships/ heterogeneity in the relationship between the dependent and explanatory variables across EAs at different spatial scale. In our study we use the corrected Akaike Information Criteria and adjusted R-squared for model comparison of OLS, GWR and MGWR model. A model with the lowest AICc value and a higher adjusted R-squared value was considered as the best-fitted model for the data.

### Ethical consideration

Since we use secondary data source that is national Ethiopian demographic health survey0f 2016 and the data set found in **www.DHSprogram.com** web site. Therefore, no need of approval to institutional review board (ethics committee). Regarding participant consent, since it is national representative data found in the above data set, It is waived and had no harm on study participates rather it is useful for decision making for intervention to prevent morbidity and mortality.

## Result

### Description of results

A total weighted sample of 10,127 under five children was included. Of these, more than half 5560 (51.8%) of the children were males. More than half of children 6241(58.2%) the children aged 25–59 month, the majority 4536(44.05%) of the children were in the Oromia region, small number of children 25 (0.25) were found in Gambela and Harari region and 9397 (88.9%) were from the rural areas. About wealth index 5015 (46.8%), and 5708 (53.1%) of the mothers fall within the poor and rich household index quintiles respectively. Nearly half of the mothers 5633 (55.6%) were aged 15–29 years, and 7089 (66.1) of the mothers no education. Regarding children's nutritional status, about 37.8%, 23.5%, and 20% of the children were stunted, underweight and wasted, respectively (**Table 1**).

## Prevalence of childhood febrile illness among under-five children in Ethiopia, 2016

The prevalence of febrile illness among under-five children was 13.6% (95% CI: 12.6%, 14 .7%). The highest prevalence of febrile illness among under-five children was observed in Oromia (14.9%%), Somalia (14.34%), and SNNPR (11.8%), Afar (10.05%), Tigray (9.56%), Amhara (8.9%), Benshangul-Gumuz (8.38%) regions Whereas, the lowest prevalence of febrile illness was observed in Addis Ababa (4.3%), Dire Dawa (5.04%), and Harari (5.78%), Gambela (6.70%) regions (**Fig 1**).

### Factors associated with febrile illness among under-five children

**Random effect analysis results.** In the null model, the ICC value was 15.2% (95% CI: 11.7%, 18.5%), indicated that about 15.2% of the overall variability in childhood febrile illness was explained by the between cluster variation while the remaining 84.8% was attributed to the individual-level variation. In addition to, the Likelihood Ratio test was (LR test vs. logistic model: $X^2(01)$ = 179.68, p< 0.0001), which showed that the mixed-effect models were the best-fitted model for this data compared to the standard model.

**Fixed effect analysis results.** After we fitted the four models model-4 was selected as the best model that had small AIc and high log likelihood ratio. We select 26 variables and 6 variables were removed due to the p-value > 0.2 then 20 variables was selected and fitted in (**Table 2**). We ran four models null model (model-I), individual level factors (model-II), Community level factors (Model-III) and both individual and community level factors (Model-IV). Then we compare the models by AIC and LL Finally model-IV the small AIC value with large LL value was selected (**Table 3**).

In the multivariable multilevel log binomial regression; age of children, maternal age, and number of children at home, diarrhea, acute respiratory infection and stunting were significantly associated with febrile illness among under-five children.

**Table 1. Socio-demographic characteristics of the study participants in Ethiopia, 2016.**

| Variable | Weighed frequency | Percentage |
|---|---|---|
| Sex of child | | |
| Male | 5560 | 51.8% |
| Female | 5163 | 48.2% |
| Age of child | | |
| 0–6 months | 1336 | 12.5% |
| 7–24 months | 3146 | 29.3% |
| 25–59 months | 6241 | 58.2% |
| Residence | | |
| Urban | 1176 | 11.1% |
| Rural | 9397 | 88.9% |
| Maternal educational status | | |
| No education | 7021 | 66.4% |
| Primary education | 2813 | 26.6% |
| Secondary education | 480 | 4.5% |
| Higher education | 259 | 2.5% |
| Wasting status | | |
| No wasting | 8422 | 89.8% |
| Wasting | 949 | 10.2% |
| Under weight | | |
| Normal weight | 7166 | 76.4% |
| Under weight | 2214 | 23.6% |
| Stunting status | | |
| No stunting | 5803 | 62% |
| Stunting | 3546 | 38% |
| Type of birth | | |
| Single | 10289 | 97.3% |
| Multiple | 283 | 2.7% |
| Place of delivery | | |
| Institutional | 2778 | 26.3% |
| Home | 7795 | 73.7% |
| ANC | | |
| No visit | 2689 | 37.4% |
| ANC Visit to three | 2217 | 30.8% |
| ANC visit to Four | 2290 | 31.8% |
| PNC | | |
| NO | 6598 | 91.6% |
| YES | 598 | 8.4% |
| Water source | | |
| Unimproved | 5914 | 56% |
| Improved | 4659 | 44% |
| Media exposure(radio and TV) | | |
| No media exposure | 7238 | 68.5% |
| Had media exposure | 3335 | 31.5% |
| BMI | | |
| Normal | 7744 | 75.2% |
| Under | 2550 | 24.8% |
| Region | | |

(*Continued*)

**Table 1.** (Continued)

| Variable | Weighed frequency | Percentage |
|---|---|---|
| Tigray | 682 | 6.6% |
| Afar | 110 | 1.06% |
| Amhara | 1954 | 18.9% |
| Oromia | 4,536 | 44.05% |
| Somalia | 493 | 4.8% |
| Benshangul-Gumuz | 112 | 1.08% |
| SNNPR | 2,205 | 20.2% |
| Gambela | 25 | 0.25 |
| Hariri | 25 | 0.25 |
| Addis-Abeba | 235 | 2.28% |
| Dire-dawa | 45.524234 | 0.4% |

The prevalence of febrile illness among children aged 7–24 months was increased by 33% (APR = 1.33, 95% CI: (1.03, 1.72)) compared to children aged 0–6 months keep other variables constant. Children who have acute respiratory infection increases febrile illness by 11.5 times (APR = 11.5, 95% CI: (9.2, 14.2) as compared to no acute respiratory infection children. The prevalence of febrile illness among children born to mothers aged 30–39 was increased by 36% (APR = 1.36 95% CI: 1.02, 1.80) compared to children born to mothers aged 15–29 years, respectively. Children who had diarrhea increases febrile illness by 4.98 times (APR = 5.3%

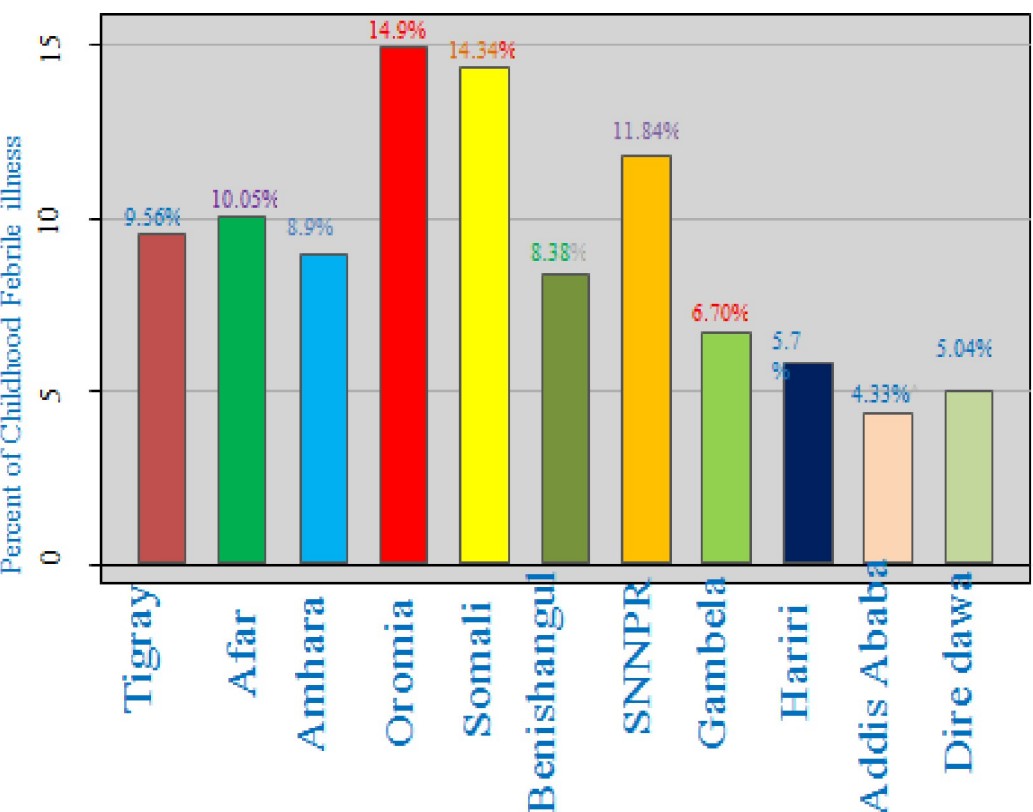

**Fig 1. Prevalence of febrile illness among under-five children across regions in Ethiopia.**

**Table 2. Bi variable and multi variable multilevel log-binomial regression for childhood febrile illness Ethiopia 2016(final model).**

| | | | | | | |
|---|---|---|---|---|---|---|
| Age of child | 0–6 month | 1040 | 153 | 1 | 1 | |
| | 7–24 months | 2250 | 516 | 1.65(1.34, 2.02) | 1.33(1.03, 1.72) | 0.02* |
| | 25–59 months | 4875 | 607 | 0.85(0.70 1.04) | 1.05(0.77, 1.42) | 0.718 |
| Stunting | No stunting | 4681 | 744 | 1 | 1 | |
| | Stunted | 2563 | 469 | 1.20(1.06 1.38) | 1.21(0.98, 1.49) | 0.05* |
| Wasting | Normal | 6455 | 1038 | 1 | 1 | |
| | Wasted | 857 | 182 | 1.34(1.10, 1.60) | 1.13(0.87, 1.48) | 0.34 |
| Underweight | Normal weight | 5523 | 878 | 1 | 1 | |
| | Underweight | 1778 | 343 | 1.28(1.109, 1.49) | 1.12(0.88, 1.43) | 0.252 |
| Birth order | Last order | 6525 | 1001 | 1 | 1 | |
| | First order | 1610 | 275 | 1.09(0.93 1.27) | 1.15(0.91, 1.44) | 0.37 |
| Type of birth | Single birth | 7962 | 1238 | 1 | 1 | |
| | Multiple birth | 173 | 38 | 1.47(1.00, 2.17) | 1.89(0.72, 4.83) | 0.19 |
| Duration of breast feeding | Ever breastfed | 4545 | 571 | 1 | 1 | |
| | Never breastfed | 325 | 37 | 0.9(0.62, 1.31) | 0.69(0.38, 1.24) | 0.23 |
| | Still breastfed | 3265 | 668 | 1.66(1.46, 1.88) | 0.87(0.69, 1.10) | 0.27 |
| Diarrhea | No | 7566 | 569 | 1 | 1 | |
| | Yes | 829 | 447 | 7.87(6.71, 9.23) | 4.98(4.09, 6.06) | 0.00 |
| ARI | No | 7794 | 778 | 1 | 1 | |
| | Yes | 341 | 498 | 15.(13, 18) | 11.4(9, 14 | 0.000* |
| Number of children at home | Up to three children | 7919 | 1236 | 1 | 1 | |
| | Four and above children | 216 | 40 | 1.42(0.97, 2.09) | 1.78(0.96, 3.3) | 0.05* |
| Vitamin A in the last 6 months | No | 4635 | 661 | 1 | 1 | |
| | Yes | 3500 | 615 | 1.25(1.09, 1.42) | 1.04(0.87, 1.24) | 0.6 |
| Educational level of mothers | No education | 5238 | 771 | 1 | 1 | |
| | Primary education | 2022 | 367 | 1.19(1.02, 1.38) | 1.09(0.88, 1.35) | 0.39 |
| | Secondary education | 557 | 93 | 1.12(0.85, 1.43) | 1.01(0.70, 1.46) | 0.91 |
| | Higher education | 318 | 45 | 0.94(0.65, 1.34) | 0.88(0.54, 1.43) | 0.61 |
| ANC visit | No visit | 1835 | 310 | 1 | 1 | |
| | Visit to three | 1603 | 310 | 1.15(0.95 1.39) | 0.96(0.76, 1.21) | 0.76 |
| | Visit four and above | 1980 | 411 | 1.23(1.02, 1.48) | 1.05(0.82, 1.34) | 0.68 |
| PNC visit | No | 4916 | 914 | 1 | 1 | |
| | Yes | 502 | 117 | 1.23(0.97, 1.55) | 0.88(0.66, 1.17) | 0.39 |
| Age of mother | 15–29 | 4359 | 701 | 1 | | |
| | 30–39 | 3118 | 486 | 0.96(0.84, 1.094) | 1.36(1.02, 1.80) | 0.03* |
| | 40–49 | 658 | 89 | 0.85(0.66, 1.090) | 0.75(0.53, 1.06) | 0.108 |
| Community Media exposure | No media exposure | 5454 | 824 | 1 | 1 | |
| | Had media exposure | 2681 | 452 | 1.08(0.94, 1.247) | 1.19(0.93, 1.51) | 0.153 |
| Residence | Urban | 1572 | 248 | 1 | 1 | |
| | Rural | 6563 | 1028 | 0.98(0.79, 1.22) | 0.77(0.47, 1.26) | 0.663 |
| BMI | Normal weight | 5581 | 855 | 1 | 1 | |
| | Under weight | 229 | 393 | 1.16(1.01 1.34) | 1.11(0.86, 1.44) | 0.42 |
| Cluster altitude | High | 2273 | 5862 | 1 | 1 | |
| | Low | 367 | 909 | .97(0.80, 1.18) | 1.05(0.74, 1.47) | 0.78 |

CI: (4.09, 6.06)) as compared to who had no diarrhea. Being the 4th and above number of children increases the prevalence of febrile illness by 1.78 times (APR = 1.78, 95% CI: 0.96, 3.3) than first to third number of births. Stunting children had 1.21 times (APR = 1.21; 95% CI:

**Table 3. Comparison of multi- level model for childhood febrile illness Ethiopia 2016.**

| Model | Observation | LL | Df | AIC | BIC |
|---|---|---|---|---|---|
| Model-I | 9411 | -3642.998 | 2 | 7289.997 | 7304.296 |
| Model-II | 5868 | -2096.018 | 15 | 4242.037 | 4408.969 |
| Model-III | 9411 | -3595.082 | 25 | 7220.163 | 7327.408 |
| Model-IV | 5868 | -2075.152 | 38 | 4226.304 | 4480.041 |

(0.98, 1.49) a higher prevalence of febrile illness compared to normal children, keep other variables constant in (**Table 2**).

**Spatial distribution of febrile illness among under-five children.** The highest prevalence of febrile illness among under-five children was observed in Tigray, North SNNPR, Dire Dawa, Benshangul Gumuz, West afar and East Amhara regions in (**Fig 2**). The spatial

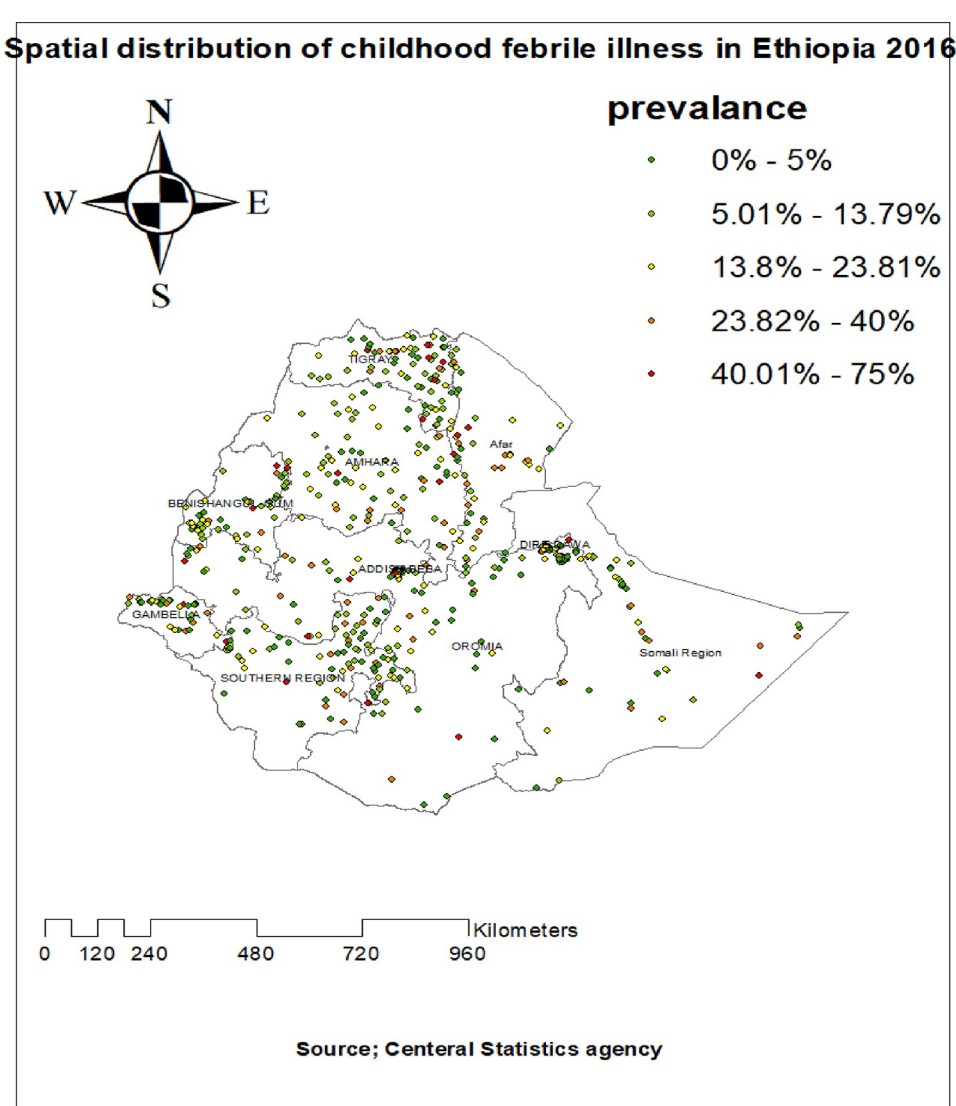

**Fig 2. Spatial distribution of febrile illness among under-five children in Ethiopia 2016.**

distribution of febrile illness among under five children showed clustered, significant spatial variation across the country with a global Moran's I value of 0.148, Z-score 8.440851 and p-value < 0.005 in (**Fig 3**). The statistically significant hotspot areas of febrile illness among under five children were identified in the Entire Tigray, Southeast of Amhara, North part of SNPPR, South afar, and Northeast Addis Abeba regions. While significant cold spot areas were detected in the central and southwest Amhara, Somalia, Dire Dawa, Benishangul-Gumuz, and regions (**Fig 4**).

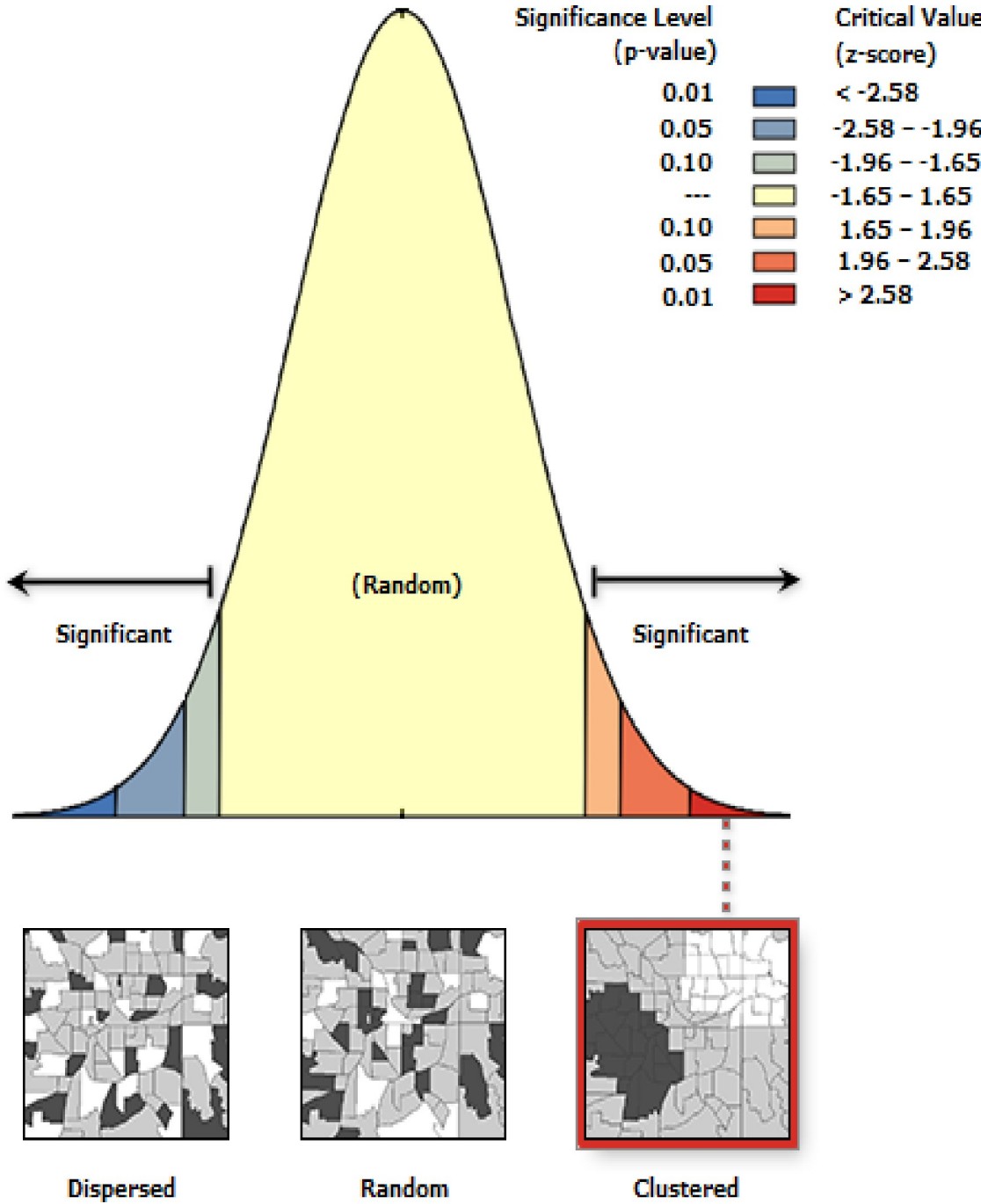

**Fig 3. Global spatial autocorrelation analysis of among under-five children in Ethiopia, 2016.**

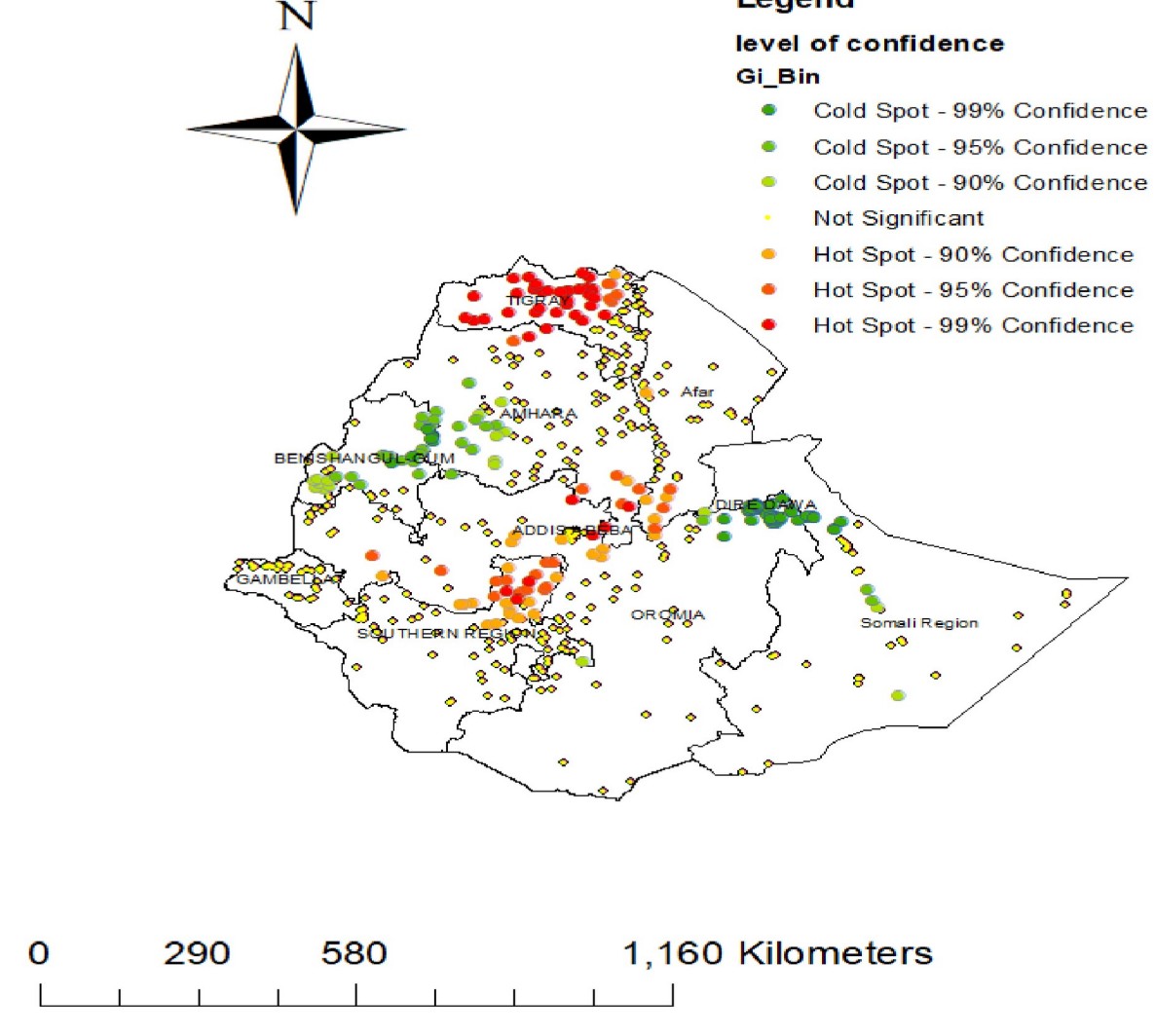

**Fig 4. The Getis Ord Gi statistical analysis of hot spots of febrile illness among under-five children in Ethiopia, 2016.**

**The global ordinary least square regression analysis results.** In the Global ordinary least square regression analysis, the model explained about 33% (adjusted R2 = 0.33) of the variation in child hood febrile illness among under five children with Akaki information criteria = -957. The Joint F-statistics and Wald statistics were significant (p<0.05), which proves that the model was statistically significant. The OLS model was calibrated to diagnose multi-collinearity among the independent variables and the mean VIF was less than 10 for each variables. The spatial distribution of residuals was not normally distributed as the Jarque-Bera statistics were significant when p- value < 0.0001. In our study the Koenker statistics were statistically significant, indicates that the relationship between the independent variables and the dependent

**Table 4. The ordinary least square regression analysis result for childhood febrile illness Ethiopia 2016.**

| Variable | Coefficient | Robust standard error | Robust t-statistics | Robust probability | VIF |
|---|---|---|---|---|---|
| Intercept | 0.047 | 0.019 | 2.36 | 0.018* | —————— |
| Proportion of children 7–24 months | -0.025 | 0.0405 | -0.2625 | 0.532 | 1.057 |
| Proportion of women who ANC follow up | -0.0011 | -0.00065 | -1.7447 | 0.08 | 1.098 |
| Proportion of mothers who had primary education | -0.028 | 0.023 | -1.2001 | 0.230 | 1.06 |
| Proportion of children who had first birth order | 0.0745 | 0.0345 | 2.159 | 0.0312* | 1.25 |
| Proportion of women who PNC follow up | 0.0810 | 0.0388 | 2.088 | 0.0372* | 1.09 |
| Proportion of mothers age 30–39 years | 0.0047 | 0.0274 | 0.1722 | 0.8632 | 1.08 |
| Proportion of children who had diarrhea | 0.2780 | 0.0441 | 6.30 | 0.00000* | 1.09 |
| Proportion of underweight children | 0.0062 | 0.0297 | 0.2093 | 0.8342 | 1.76 |
| Proportion of children who had acute respiratory infection | 0.5687 | 0.0554 | 10.256 | 0.000000* | 1.15 |
| Proportion of stunted children | -0.0012 | 0.0276- | -0.0435 | 0.9652 | 1.76 |
| **Number of observations** | **640** | **Adjusted R-squared** | | **0.336** | |
| Joint F-statistics | 33.35 | Prob(>F), (10, 629) degree of freedom | | 0.000000* | |
| Joint Wald statistics | 225.6 | Prob (> chi-squared), (10) degree of freedom | | 0.000000* | |
| Koenker (BP) statistics | 80.98 | Prob (> chi-squared), (10) degree of freedom | | 0.000000* | |
| Jarque–Bera statistics | 242.35 | Prob (> chi-squared), (2) degree of freedom | | 0.000000* | |

variable was non-stationary or heterogeneous across the study areas. This indicates that GWR should be applied as it assumes the spatial heterogeneity of the relationship between independent and dependent variables across space. Since both joint Wald and joint f-statics were significant, we were used robust probability to select the statistically significant variables for GWR analysis.

The proportion of women who had ANC follow up, the proportion of children who had first birth order, the proportion of women who PNC, the proportion of children who had acute respiratory infection, and the proportion of children who had diarrhea were significantly associated with the percentage of febrile illness among children under-five children in the OLS model (**Table 4**).

**The geographically weighted regression analysis result.** The Geographically weighted regression analysis showed that there was a significant improvement over global ordinary least square regression. The AICc value decreased from -957 for the OLS model to -972 for the GWR model. The difference was 12 implied that the GWR best explains the spatial heterogeneity of childhood febrile illness among under five children. Besides, the adjusted R2 was 0.42, the model's ability to explain childhood febrile illness among under five children has been improved by using GWR since the adjusted R2 was 0.42, indicates that GWR improved the model explaining the power of the OLS model by about 9% (**Table 5**).

**The multi-scale geographically weighted regression analysis result.** The Multi-Scale Geographically weighted regression analysis showed that there was no significant improvement over global ordinary least square regression and GWR. The value increased from -972 for the GWR model to 1478 in MGWR model. Besides, the adjusted R2 was 0.42which is the same as GWR. As a result GWR was better than OLS and MGWR models in (**Table 5**).

**Table 5. Model comparison of OLS, GWR and MGWR model.**

| Model Comparison parameter | OLS model | GWR Model | MGWR model |
|---|---|---|---|
| AICc | -957 | -972 | 1478 |
| Adjusted R-squared | 0.33 | 0.42 | 0.42 |

In the Geographically weighted regression analysis, the proportion of women who had Postnatal care, the proportion of children who had diarrhea, the proportion of children who had acute respiratory infection, the proportion of being 1st birth order, were considered as statistically significant explanatory variables that influence the outcome variable.

The proportion of children who had diarrhea had a positive relationship with the proportion of childhood febrile illness among children aged 6–59 months. As the proportion of children who had diarrhea increased, the percentage of febrile illness among children aged 6–59 months increased in the entire Tigray, Benishangul-gumuz, west part of Oromia, North part of SNNPR and Somalia regions. The geographic area with red colored points indicates the highest coefficient of the proportion of children diarrhea (Fig 5). Surprisingly the proportion of mothers PNC was significantly associated with the increased risk of febrile illness among

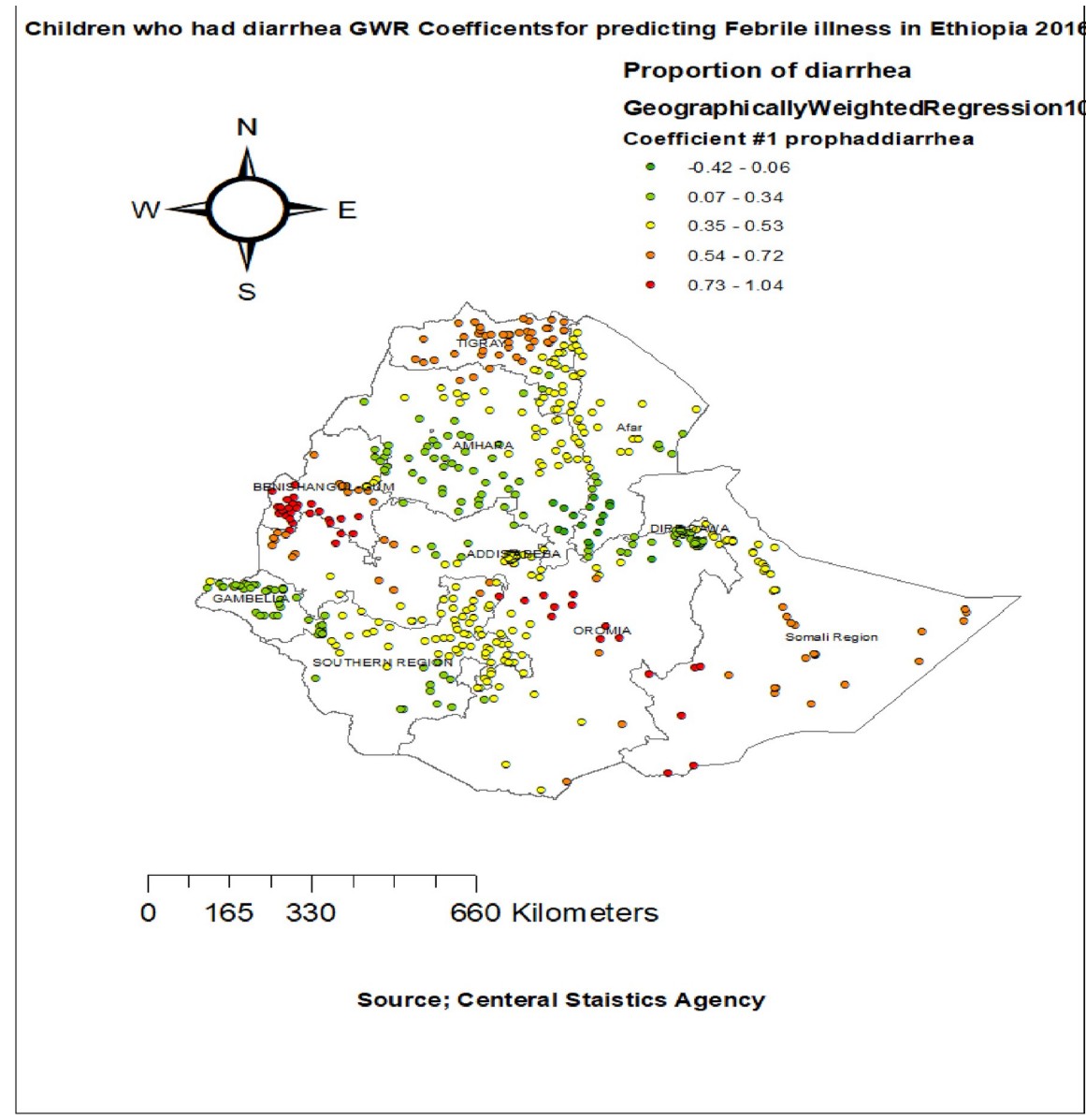

**Fig 5. Children who had diarrhea GWR coefficients for predicting febrile illness among under five children in Ethiopia, 2016.**

children aged 6–59 months, with the highest effect of PNC observed in entire Somalia region, north east Amhara, South afar, and SNNPR regions (**Fig 6**).

The proportion of children who had acute respiratory infection showed strong and positively associated with increased risk of febrile illness among children aged 6–59 months in entire Tigray region, Dire-Dawa, South Afar, north SNNPR and north Somali regions (**Fig 7**). The proportion of children being first birth order had a significant positive association with febrile illness among children aged 6–59 months in east Tigray, entire Amhara, Gambela and north Oromia and SNNPR regions (**Fig 8**).

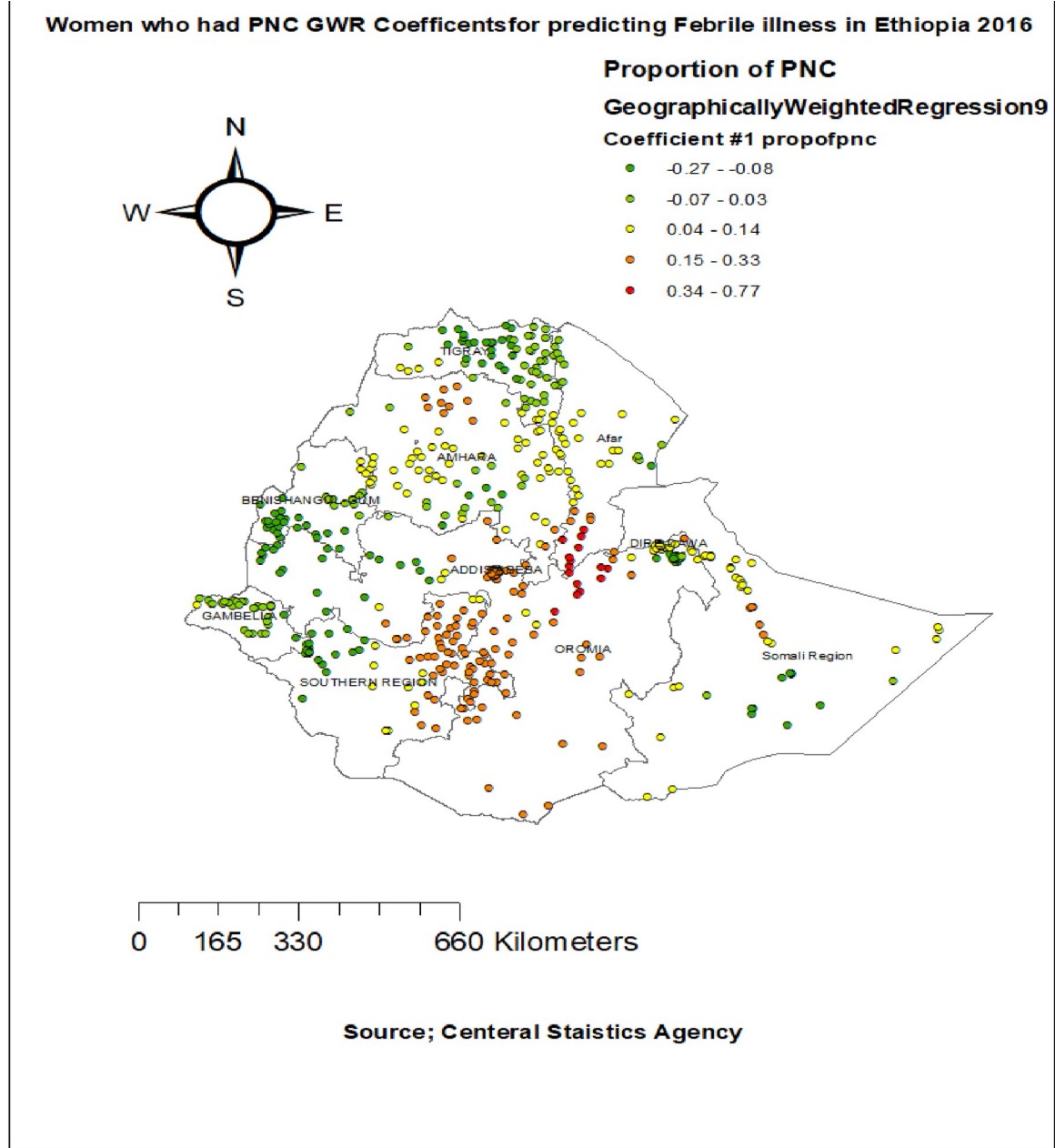

**Fig 6. Women who had PNC months GWR coefficients for predicting febrile illness among under five children in Ethiopia, 2016.**

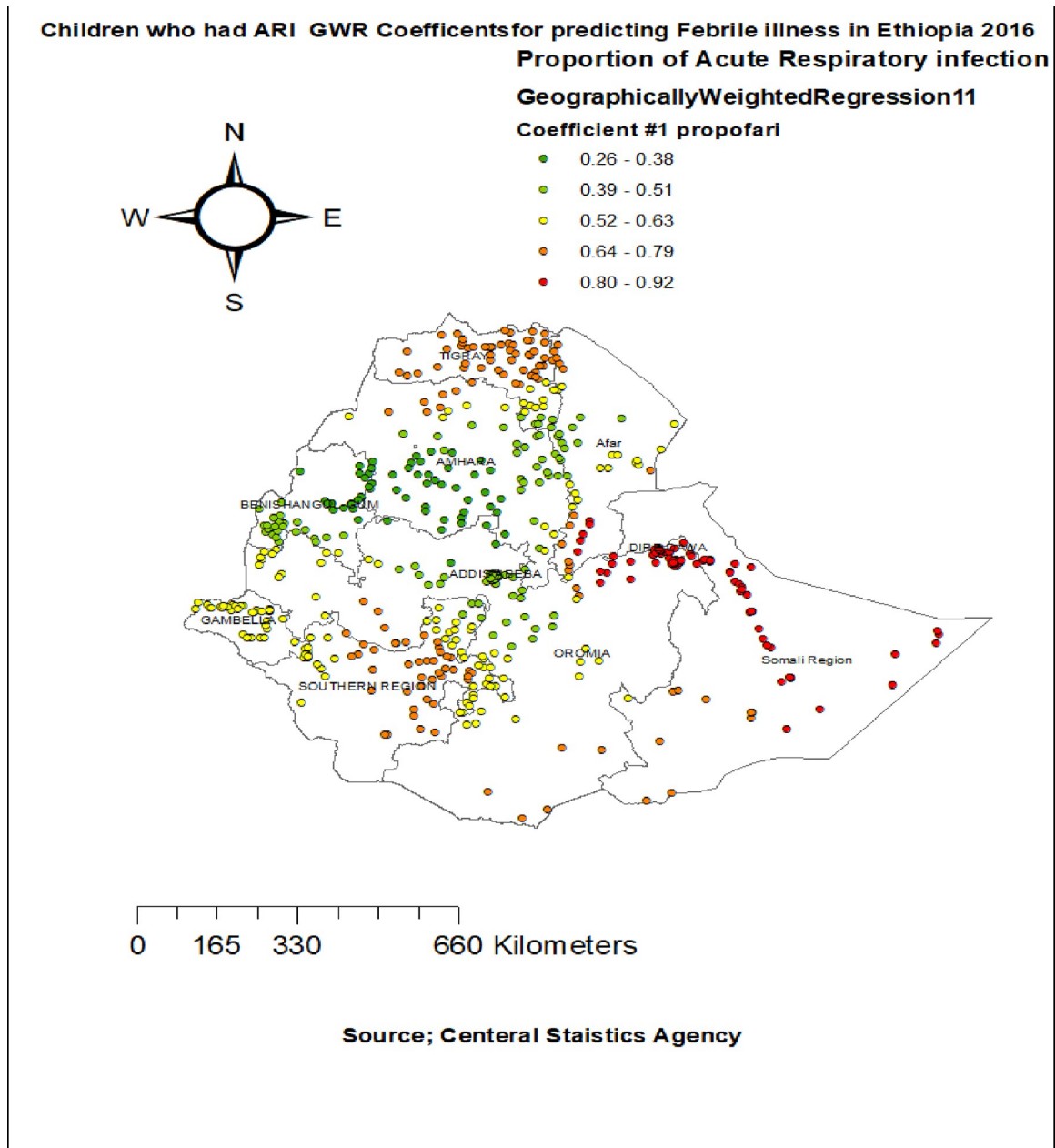

**Fig 7. Children who had ARI GWR coefficients for predicting febrile illness among under -five children in Ethiopia, 2016.**

## Discussion

Febrile illness among under-five children remains public health problem in Ethiopia. In this study, the prevalence of febrile illness among under-five children in Ethiopia was 13.6% ranged from 4.3% in the Addis Abeba to 14.9% in the Oromia region. This is consistent finding in Zanzibar [31], Nigeria [32], but lower than Sub-Sahara countries(Ghana, Kenya, Sierra Leone) [20]. The possible reason is might be different socio demographic characteristics of the study participates [33], different health system and infrastructure [34], and different geographical location [35] that are susceptible to febrile illness also another factor that have different prevalence of febrile illness. Most care givers or mothers do not going to health facilities for the

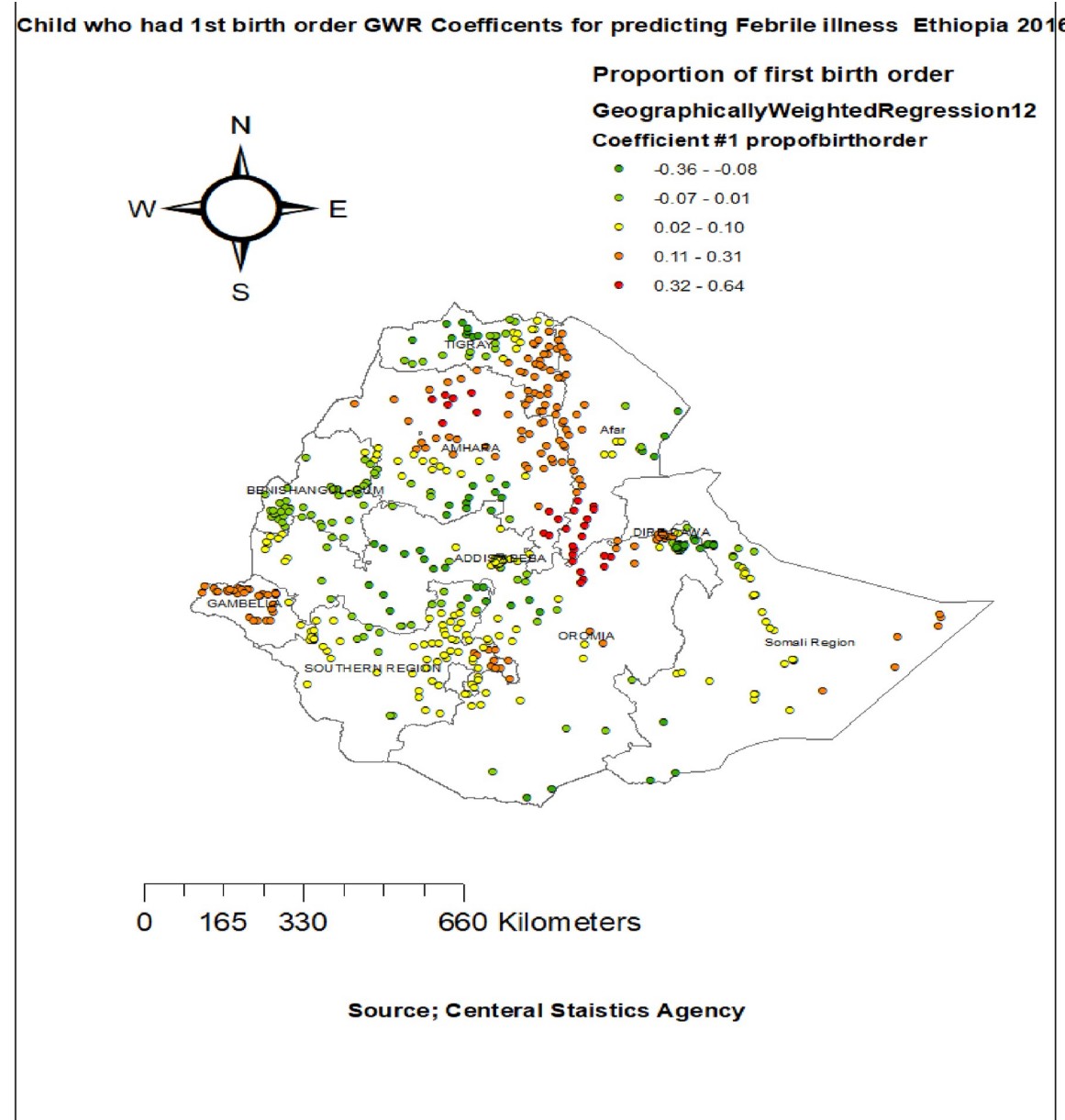

**Fig 8. Children who had first birth order GWR coefficients for predicting febrile illness among under -five children in Ethiopia, 2016.**

treatment of her children by febrile illness due to lack of money, did not see any benefit [36] which leads to underreported of febrile illness.

Even though low prevalence of childhood febrile illness in Ethiopia, the spatial distribution of febrile illness among under five children in Ethiopia was non-random and the hotspot areas of Febrile illness were identified in the entire Tigray, Southeast of Amhara, North part of SNPPR, South afar, and Northeast Addis Abeba regions. Most clusters found in Tigray and SNNPR regions, Therefore, health education on health seeking behavior of women to health facilities [37], appropriate laboratory investigation and provide supportive management and treatment [38] is essential to reduce child morbidity and mortality.

In GWR analysis the proportion of children who had diarrhea were had a positive relationship with the proportion of febrile illness among under-five children. As the proportion of children who had diarrhea increased, the percentage of febrile illness among under five children increased in the entire Tigray and Benishangul-gumuz region This finding consistent with in Peru [39] previous studies in Ethiopia [22, 40]. This might be suppression of immunity, lack of potable water supply and sanitation in Ethiopia national coverage of 24% [41], lack of appropriate feeding practices for children leads to infectious diseases. Then the government of Ethiopia improving clean water supply coverage and sanitation, provide treatment for sick children like Zinc and ORS, providing clean foods for children with balanced diet in their age category is important.

Surprisingly women who had PNC, increases the percentage of febrile illness among under five children, in Somalia region and south east Amhara. This finding is contradicted with the previous study [42]. The possible reason is during post natal care follow up at health facilities, poor diseases prevention mechanisms [43], diseases transmitted from infected person to mothers and neonate during PNC by bite of vectors like mosquito [44], by contact with infected person or health professionals that provide post natal care services like common cold. Even though providing PNC is important for reducing maternal and child mortality [45], During PNC, health care providers should use diseases prevention mechanism like face mask, Social distancing, use of bed nets.

The proportion of children, who had acute respiratory infection, strongly associated with percentage of febrile illness among under five children age in entire Tigray region, Dire dawa, South Afar, north SNNPR and north Somali regions. This finding in line with previous studies reported in Ethiopia [22], Africa [46], Australia [47]. The possible reason is during respiratory infection there is metabolic change that response to the diseases by thermoregulation from CNS [48], immune compromised that exposed to febrile illness. As a result, early treatment of acute respiratory infection prevents child hood febrile illness.

The other explanatory variable is proportion of children who had first birth order were significantly associated with the percentage of febrile illness among under five children East Tigray, entire Amhara, Gambela and north Oromia and SNNPR regions. This similar that of previous research in Ethiopia [42], New York [49], This might be lack of experience of mother for caring her child, lack of nutrition from his/her mother during prim-gravidity.

In the multivariable multi-level log binomial regression; age of children, maternal age and number of children at home, diarrhea, acute respiratory infection and stunting were significantly associated with febrile illness among under-five children.

Being child in the age of 7–24 months increases the risk of febrile illness as compared to 0–6 months of age. This finding similar that of a study done at Ethiopia [22], Nigeria [15]. The possible reason could be due to complementary feeding is initiated after 6 months of birth and during this, a child is exposed to contaminated food and mal absorption syndrome, this could results immune suppression. In addition, travelling more risk areas from his/her mother, feeding contaminated foods by him/herself, drinking unclean water by their own self, decreases care from mothers due to mothers work overload [50]. Therefore, improving appropriate care to children was expected from mothers or care givers, vaccination for preventing infectious diseases like pneumonia, use of bed net for malaria cause febrile illness.

A child who is stunting was more likely to be febrile illness than their counterparts. This is in line with previous studies reported in Ethiopia [22]. This could be due to low intake of foods and diminished nutrient absorption caused by changes in the gastrointestinal epithelium in malnourished individuals contribute towards the development of febrile illness immune suppression, does not have necessary nutrients in his/her body, prone to other comorbidities

and febrile illness. Multi sectorial collaboration was necessary to prevent malnutrition in order to prevent child morbidity and mortality by febrile illness.

The other explanatory variable is the number of children at home, As four and above children found at home increases the risk of febrile illness as compared to below three children at home. This finding is consistent with previous studies reported in Ethiopia [22]. The possible reason might be increases the necessary basic foods, does not filled cares for all children, poverty may is there, lack of appropriate feeding and care practice to numerous children.

The final explanatory variable was mother's age. A child born to a mother aged 30–39 years had increases prevalence of febrile illness compared to a child born to a mother aged 15–29 years. These findings were in line with previous studies reported in Nigeria [51]. The reason behind might be due to decreases energy for caring her child, the mothers traveling more frequently may go to malaria endemic areas, work hard to increase income to their children, then the child may not get appropriate care, prone to infection leads child hood febrile illness.

## Strength and limitation of the study

Our study has strength and limitations. The EDHS data was nationwide in Ethiopia, which is representative of the whole population that is generalizable. This study is new by geographically weighted regression analysis childhood febrile illness in Ethiopia, base line information for other researchers and policy makers. The limitation of this study was since EDHS data rely on information gathered from caregivers, and responses may be colored by widely varying beliefs about perceptions of symptoms may bias our findings. In addition, the use of a 2-week recall period for illness history has been shown to lead to an underestimation of disease incidence, and may have affected our findings.

## Conclusion and recommendation

Febrile illness among under-five children remains public health problem in Ethiopia. In this study, the prevalence of febrile illness among under five children disproportional located across regions in Ethiopia. The hotspot areas of febrile illness were identified in the entire Tigray, Southeast of Amhara, South afar, Northeast Addis Abeba and North part of SNNPR regions. In the multivariable mixed-effect logistic regression; age of children, maternal age, birth order, diarrhea, acute respiratory infection, stunting, and number of children were significantly associated with febrile illness among under-five children. In the Geographically weighted regression analysis, the proportion of women who had postnatal care, the proportion of children who had diarrhea, the proportion of children who had acute respiratory infection, the proportion of being 1st birth order, were considered as statistically significant explanatory variables. Therefore, the government implement strategies for preventing childhood febrile illness at hotspot area of the country by providing health education on health seeking behavior of women to her ill child early going to health facilities, improved hygiene and sanitation for children by providing appropriate care, use of insecticide treated bed net appropriately, early diagnosis and treatment of children to prevent child morbidity and mortality. The detailed research report and map of childhood febrile illness and its predictors in Ethiopia could assist health program planners, NGO and policy makers to design targeted public health interventions for febrile illness.

## Supporting information

**S1 Data.**
(DTA)

## Acknowledgments

We greatly acknowledge MEASURE DHS for granting access to the EDHS data sets.

## Author Contributions

**Conceptualization:** Negalgn Byadgie Gelaw, Getayeneh Antehunegn Tessema, Kassahun Alemu Gelaye, Zemenu Tadesse Tessema, Tigist Andargie Ferede, Abebe W/Selassie Tewelde.

**Data curation:** Negalgn Byadgie Gelaw, Getayeneh Antehunegn Tessema, Kassahun Alemu Gelaye, Zemenu Tadesse Tessema, Tigist Andargie Ferede, Abebe W/Selassie Tewelde.

**Formal analysis:** Negalgn Byadgie Gelaw, Getayeneh Antehunegn Tessema, Kassahun Alemu Gelaye, Zemenu Tadesse Tessema, Tigist Andargie Ferede, Abebe W/Selassie Tewelde.

**Investigation:** Negalgn Byadgie Gelaw.

**Methodology:** Negalgn Byadgie Gelaw, Getayeneh Antehunegn Tessema, Kassahun Alemu Gelaye, Zemenu Tadesse Tessema, Tigist Andargie Ferede, Abebe W/Selassie Tewelde.

**Resources:** Negalgn Byadgie Gelaw, Tigist Andargie Ferede.

**Software:** Negalgn Byadgie Gelaw, Getayeneh Antehunegn Tessema, Kassahun Alemu Gelaye, Zemenu Tadesse Tessema, Tigist Andargie Ferede, Abebe W/Selassie Tewelde.

**Supervision:** Negalgn Byadgie Gelaw, Getayeneh Antehunegn Tessema, Kassahun Alemu Gelaye, Zemenu Tadesse Tessema.

**Validation:** Negalgn Byadgie Gelaw, Getayeneh Antehunegn Tessema, Kassahun Alemu Gelaye, Zemenu Tadesse Tessema, Tigist Andargie Ferede, Abebe W/Selassie Tewelde.

**Visualization:** Negalgn Byadgie Gelaw, Getayeneh Antehunegn Tessema, Kassahun Alemu Gelaye, Zemenu Tadesse Tessema, Tigist Andargie Ferede, Abebe W/Selassie Tewelde.

**Writing – original draft:** Negalgn Byadgie Gelaw, Tigist Andargie Ferede, Abebe W/Selassie Tewelde.

**Writing – review & editing:** Negalgn Byadgie Gelaw, Tigist Andargie Ferede, Abebe W/Selassie Tewelde.

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
