## [Decision Letter · Decision Letter 0]

20 Jul 2022

PONE-D-22-11036Exploring the Spatial Variation and Associated Factors of Childhood Febrile Illness Among Under-Five Children In Ethiopia: A Geographically Weighted Regression AnalysisPLOS ONE

Dear Dr. Negalgn Byadgie Gelaw,

Thank you for submitting your manuscript to PLOS ONE. After careful consideration, we feel that it has merit but does not fully meet PLOS ONE’s publication criteria as it currently stands. Therefore, we invite you to submit a revised version of the manuscript that addresses the points raised during the review process.It is also recommended to do the extensive editing and present the table and graphs in an appropriate form. Please submit your revised manuscript by 31st Aug 2022 . If you will need more time than this to complete your revisions, please reply to this message or contact the journal office at plosone@plos.org. Please include the following items when submitting your revised manuscript:A rebuttal letter that responds to each point raised by the academic editor and reviewer(s). You should upload this letter as a separate file labeled 'Response to Reviewers'.A marked-up copy of your manuscript that highlights changes made to the original version. You should upload this as a separate file labeled 'Revised Manuscript with Track Changes'.An unmarked version of your revised paper without tracked changes. You should upload this as a separate file labeled 'Manuscript'.

We look forward to receiving your revised manuscript.

Kind regards,

Jayanta Kumar Bora, PhD

Academic Editor

PLOS ONE

Journal Requirements:

2. We note that Figure 2, 4, 6, 7, 8 in your submission contain [map/satellite] images which may be copyrighted. All PLOS content is published under the Creative Commons Attribution License (CC BY 4.0), which means that the manuscript, images, and Supporting Information files will be freely available online, and any third party is permitted to access, download, copy, distribute, and use these materials in any way, even commercially, with proper attribution. For these reasons, we cannot publish previously copyrighted maps or satellite images created using proprietary data, such as Google software (Google Maps, Street View, and Earth). For more information, see our copyright guidelines: http://journals.plos.org/plosone/s/licenses-and-copyright.

1. You may seek permission from the original copyright holder of Figure 2, 4, 6, 7, 8 to publish the content specifically under the CC BY 4.0 license.  

- https://journals.plos.org/plosntds/article?id=10.1371%2Fjournal.pntd.0004040

- https://academic.oup.com/jpids/article/5/2/190/2580362?login=false

- https://link.springer.com/article/10.1186/s41182-020-00252-5?code=8e3d172f-b365-4660-93d4-49e83301f774&error=cookies_not_supported

- https://www.gov.uk/research-for-development-outputs/find-acute-febrile-syndrome-strategy

- https://pubmed.ncbi.nlm.nih.gov/1353192/

In your revision ensure you cite all your sources (including your own works), and quote or rephrase any duplicated text outside the methods section. Further consideration is dependent on these concerns being addressed

Reviewers' comments:

Reviewer's Responses to Questions

**Comments to the Author**

1. Is the manuscript technically sound, and do the data support the conclusions?

Reviewer #1: Yes

Reviewer #2: Yes

2. Has the statistical analysis been performed appropriately and rigorously? 

Reviewer #1: Yes

Reviewer #2: Yes

3. Have the authors made all data underlying the findings in their manuscript fully available?

Reviewer #1: Yes

Reviewer #2: No

4. Is the manuscript presented in an intelligible fashion and written in standard English?

Reviewer #1: Yes

Reviewer #2: Yes

5. Review Comments to the Author

Reviewer #1: This is an interesting study and the paper is generally well written and structured. The author has extensively explored the spatial variation and associated factors of childhood febrile illness among under-five children in Ethiopia. The literature review was thorough, the methodology was painstakingly thorough, and incorporated the use of sufficient numbers of samples. Overall, the information presented represents valuable information regarding the geographical variation of childhood febrile illness.

Reviewer #2: 1. remove repeated words (line 31)

2. if so why your findings indicates high% in oromia region? correlate with figure 8.(line 32)

3. correct the word are you want to say woreda/district??? (line 93)

4. is (10127) data was your total sample size or for only one cluster? line 102

5. sample size(N) should be clearly indicated with the preferred statistical method line 102

6. PLOS authors have the option to publish the peer review history of their article (what does this mean?). If published, this will include your full peer review and any attached files.

Reviewer #1: **Yes: **Ankita Srivastava

Reviewer #2: No

---

## [Author Response · Author response to Decision Letter 0]

16 Sep 2022

I was responded to reviewers and editors point by point response to fulfill PLOSONE criteria " Responses to Reviewers"

For manuscript sent back to author I will response on "enter comments" and revised manuscript with new track changes

 #######Dear editors/reviewers#######

Here is the responses for constructive comments

#Regarding minor occurrence of overlapping text with the following previous publication

our study focus on the spatial variation and associated factors of childhood febrile illness in Ethiopia by EDHS 2016 with multi level log binomial analysis and geographically weighted regression analysis the other studies different from us these are

1. The first article is https://journals.plos.org/plosntds/article?id=10.1371%2Fjournal.pntd.0004040 (Vitamin A supplementation and child survival) was discussed about vit.A supplementation and child survival witch is different from us discussed about vit.a supplementation on child mortality

2.The second article is FIND acute febrile syndrome strategy witch is find common syndrome approach for acute febrile

 illness to diagnose the diseases and to provide treatment, which is different from us

3. The third article is Geographical disparities and determinants of "childhood diarrheal illness" in Ethiopia which is different from us exploring the spatial variation and associated factors of "childhood febrile illness" plus our study is on geographically weighted regression and multi scale geographical weighted regression analysis, which is totally different by outcome and analysis

4. the 4th article on Challenges in the Etiology and Diagnosis of Acute Febrile Illness in Children in Low- and Middle-Income Countries which different from us because it discusses on etiologic bacteria of febrile illness

5. The fifth article on Estimating the Burden of Febrile Illnesses and its challenges worldwide which is different from us because it discusses extrapolation of the incidence and else

#regarding supporting information -no supplementary file except figures

# regarding Table-1 and table -3 , we will correct it in "Revised Manuscript with new track changes"

 Thank you for your kind feedback!!!

---

## [Decision Letter · Decision Letter 1]

31 Oct 2022

Exploring the Spatial Variation and Associated Factors of Childhood Febrile Illness Among Under-Five Children In Ethiopia: A Geographically Weighted Regression Analysis

PONE-D-22-11036R1

Dear Dr. Negalgn Byadgie Gelaw,

We’re pleased to inform you that your manuscript has been judged scientifically suitable for publication and will be formally accepted for publication once it meets all outstanding technical requirements.Please check and correct all the tables and map formats according to the PlosOne criteria.

Kind regards,

Jayanta Kumar Bora, PhD

Academic Editor

PLOS ONE

Additional Editor Comments (optional):

Reviewers' comments:

Reviewer's Responses to Questions

**Comments to the Author**

1. If the authors have adequately addressed your comments raised in a previous round of review and you feel that this manuscript is now acceptable for publication, you may indicate that here to bypass the “Comments to the Author” section, enter your conflict of interest statement in the “Confidential to Editor” section, and submit your "Accept" recommendation.

Reviewer #2: All comments have been addressed

2. Is the manuscript technically sound, and do the data support the conclusions?

Reviewer #2: Yes

3. Has the statistical analysis been performed appropriately and rigorously? 

Reviewer #2: Yes

4. Have the authors made all data underlying the findings in their manuscript fully available?

Reviewer #2: Yes

5. Is the manuscript presented in an intelligible fashion and written in standard English?

Reviewer #2: No

6. Review Comments to the Author

Reviewer #2: I have checked my previously commented to the authors and is ok if accepted! All comments have been fully addressed by the authors.

7. PLOS authors have the option to publish the peer review history of their article (what does this mean?). If published, this will include your full peer review and any attached files.

Reviewer #2: No

---

## [Editor Report · Acceptance letter]

19 Dec 2022

PONE-D-22-11036R1 

Exploring the Spatial Variation and Associated Factors of Childhood Febrile Illness among Under-Five Children In Ethiopia:  Geographically Weighted Regression Analysis 

Dear Dr. Gelaw:

I'm pleased to inform you that your manuscript has been deemed suitable for publication in PLOS ONE. Congratulations! Your manuscript is now with our production department. 

Kind regards, 

on behalf of

Dr. Jayanta Kumar Bora 

Academic Editor

PLOS ONE